

# Design and validation of a next generation sequencing assay for hereditary *BRCA1* and *BRCA2* mutation testing

Hyunseok P. Kang, Jared R. Maguire, Clement S. Chu, Imran S. Haque, Henry Lai, Rebecca Mar-Heyming, Kaylene Ready, Valentina S. Vysotskaia and Eric A. Evans

Counsyl Inc., South San Francisco, CA, United States

## ABSTRACT

Hereditary breast and ovarian cancer syndrome, caused by a germline pathogenic variant in the *BRCA1* or *BRCA2* (*BRCA1/2*) genes, is characterized by an increased risk for breast, ovarian, pancreatic and other cancers. Identification of those who have a *BRCA1/2* mutation is important so that they can take advantage of genetic counseling, screening, and potentially life-saving prevention strategies. We describe the design and analytic validation of the Counsyl Inherited Cancer Screen, a next-generation-sequencing-based test to detect pathogenic variation in the *BRCA1* and *BRCA2* genes. We demonstrate that the test is capable of detecting single-nucleotide variants (SNVs), short insertions and deletions (indels), and copy-number variants (CNVs, also known as large rearrangements) with zero errors over a 114-sample validation set consisting of samples from cell lines and deidentified patient samples, including 36 samples with *BRCA1/2* pathogenic germline mutations.

## INTRODUCTION

Hereditary breast and ovarian cancer syndrome (HBOC) is associated with mutations in tumor suppressor genes *BRCA1* and *BRCA2.* Genetic analysis for individuals who are at risk for HBOC has become widely accepted. Several professional organizations and expert panels, including the National Comprehensive Cancer Network (NCCN) (*National Comprehensive Cancer Network*, *2014*), the American Society of Clinical Oncology (ASCO) (*Robson et al.*, *2010*), the American Society of Human Genetics (ASHG) (*American Society of Human Genetics*, *1994*), the American College of Medical Genetics and Genomics (ACMG) (*Hampel et al.*, *2015*), the National Society of Genetic Counselors (NSGC) (*Hampel et al.*, *2015*), the US Preventive Services Task Force (USPSTF) (*Nelson et al.*, *2014*), the Society of Gynecologic Oncologists (SGO) (*Lancaster et al.*, *2007*), and the European Society for Medical Oncology (ESMO) (*Balmaña et al.*, *2011*) have developed clinical criteria and practice guidelines for identifying individuals who may benefit from *BRCA1* or *BRCA2* mutation testing. In general, personalized risk assessment, genetic counseling, and

Corresponding author
Hyunseok P. Kang,
research@counsyl.com

**Table 1** *BRCA1* and *BRCA2* cancer risk management options and effectiveness.

| Risk management options | Effectiveness |
| --- | --- |
| Prophylactic mastectomy | Up to 90% reduction in breast cancer risk (*Hartmann et al., 1999, 2001; Meijers-Heijboer et al., 2001*) |
| Prophylactic oophorectomy | ∼50% reduction in breast cancer risk when performed premenopausally (more pronounced effect for *BRCA2* mutation carriers compared to *BRCA1*) (*Kauff et al., 2002, 2008*) up to 96% reduction in ovarian cancer risk (*Olopade & Artioli, 2004; Rebbeck et al., 2002; Rutter et al., 2003*) |
| Tamoxifen | Up to 62% reduction in breast cancer risk among *BRCA2* mutation carriers; up to 50% contralateral breast cancer risk reduction in both *BRCA1* and *BRCA2*; limited data but appears to be more effective in *BRCA2* mutation carriers compared to *BRCA1* (*King et al., 2001; Metcalfe et al., 2005; Narod et al., 2000*) |
| Oral contraceptives | Up to 50% reduction in ovarian cancer risk (*Iodice et al., 2010*) |
| Breast MRI/mammogram | No risk reduction, but earlier detection (*Kuhl et al., 2010; Sardanelli et al., 2011; Warner et al., 2011*) |
| Ovarian cancer screening (transvaginal ultrasound and serum cancer antigen 125 (CA-125)) | No risk reduction and no effect on cancer mortality (*Buys et al., 2011; Clarke-Pearson, 2009*) |

often *BRCA1/2* testing and management are recommended for individuals with a significant personal and/or family history of breast, ovarian, pancreatic and/or prostate cancer.

As suggested by various guidelines, individuals identified with *BRCA1* or *BRCA2* mutation are at significantly increased risk for breast, ovarian, prostate, pancreatic and possibly other cancers: a 12% general population risk for breast cancer rises to 50–80% for *BRCA1* mutation carriers or 40–70% for *BRCA2* mutation carriers (*Petrucelli, Daly & Feldman, 2015*). Screening for *BRCA* mutations is of great significance for breast and ovarian cancer prevention and early detection. Recommended risk-reducing options include increased screening, chemoprevention and/or prophylactic surgery (*Balmaña et al., 2011; Hampel et al., 2015; Lancaster et al., 2007; National Comprehensive Cancer Network, 2014; Nelson et al., 2014; Robson et al., 2010; American Society of Human Genetics, 1994*). Table 1 summarizes these options and their effect on cancer risks.

Genetic testing for *BRCA1/2* mutation status has the potential to offer multiple benefits. However, 20–73% of mutation carriers may not be identified by current guidelines (*Alsop et al., 2012; Brozek et al., 2012; Frank et al., 2002; Kang et al., 2014; Norquist et al., 2013*) or only meet current guidelines once they are diagnosed with ovarian cancer or early onset breast cancer, resulting in some researchers to call for more inclusive guidelines or even population screening (*Finch et al., 2014; Gabai-Kapara et al., 2014; Metcalfe et al., 2013*).

Next-generation sequencing (NGS) technologies offer higher throughput and lower per-base cost as compared to legacy approaches such as Sanger sequencing. Although researchers have described technical questions regarding analytical performance for classes of variants that are considered to be challenging for NGS (*Harismendy et al., 2009*), these refer to technologies that are several years old. Several laboratories have recently reported applying a NGS approach for diagnostic testing of mutations in the *BRCA1/2* genes or multigene panels that include the *BRCA1* and *BRCA2* genes. They performed a comparison of data analyses including independent and blind evaluation as well as power estimation of

**Table 2  Source of samples and reference data used in validation.**

| Mutation type | Test samples | Reference data |
|---|---|---|
| SNV/Indel | 41 Coriell cell line samples | 1000 Genomes Project Exomes |
| | NA12878 | Illumina Platinum Genome |
| | 15 BIC samples | BIC reference data |
| | 10 positive patient samples | Orthogonal confirmation by Sanger |
| CNV | NA12878 | Orthogonal confirmation by MLPA |
| | 15 BIC samples | Orthogonal confirmation by MLPA |
| | 13 reference lab samples | Reference lab results |
| | 25 random patient samples | Orthogonal confirmation by MLPA |
| | 9 positive patient samples | Orthogonal confirmation by MLPA |

the new NGS methodologies in comparison to Sanger sequencing and demonstrated the very high accuracy of the NGS methods (*Bosdet et al.*, *2013*; *Chong et al.*, *2014*; *Judkins et al.*, *2015*; *Lincoln et al.*, *2015*; *Strom et al.*, *2015*).

The aim of the present study was to evaluate analytical sensitivity and specificity of the Counsyl Inherited Cancer Screen (ICS), an NGS-based test for *BRCA1/2* testing. We followed ACMG guidelines for analytical validation of NGS methods and platforms (*Rehm et al.*, *2013*). The test also adheres to these guidelines for interpretation and reporting of detected variants. Here, we report the results from a validation set of 114 cell line and patient DNA samples, in which we demonstrate 100% concordance with reference data or orthogonal assays.

## MATERIALS AND METHODS

### DNA samples

The Counsyl Inherited Cancer Screen validation study was conducted by testing three classes of samples: (a) deidentified blood samples ($N = 57$), (b) deidentified paired blood and saliva samples (7 pairs), and (c) genomic DNA reference materials obtained from Coriell ($N = 57$), including the well-characterized NA12878 sample from HapMap/1000 Genomes and 15 samples from the BIC *BRCA1*/*BRCA2* Mutation Panel (Table 2 and Table S1). The protocol for this study was approved by Western Institutional Review Board (IRB number 1145639) and complied with the Health Insurance Portability and Accountability Act (HIPAA). The information associated with patient samples was deidentified in accordance with the HIPAA Privacy Rule. A waiver of informed consent was requested and approved by the IRB.

### Test design

The reportable range of the test is all coding exons of *BRCA1* (NM_007294.3) and *BRCA2* (NM_000059.3), 20 bp into the introns from intron/exon junctions, and selected intronic and untranslated regions where pathogenic variants have been reported in the literature (Table S2).

## Next generation sequencing

DNA from a patient's blood or saliva sample is isolated, quantified by a Picogreen fluorescence assay and then fragmented to 200–1,000 bp by sonication. The fragmented DNA is converted to an adapter-ligated sequencing library by end repair, A tailing, and barcoded adapter ligation; samples are multiplexed and identified by molecular barcodes. Hybrid capture-based enrichment with 40-mer oligonucleotides (Integrated DNA Technologies, Coral, IL, USA) complementary to *BRCA1/2* targeted regions is performed on these multiplexed samples. Next generation sequencing of the selected targets is performed with sequencing-by-synthesis on the Illumina HiSeq 2500 instrument to mean sequencing depth of ∼500×. All target nucleotides were required to be covered with a minimum depth of 20 reads.

## Bioinformatics processing

Generated sequence reads are aligned to the hg19 human reference genome using the BWA-MEM algorithm (*Li, 2013*), which also trims sequencing adapters. Automated statistical analysis is used to identify and genotype single-nucleotide variants (SNVs) and short insertions and deletions (indels) following methods in GATK 1.6 and FreeBayes (*Garrison & Marth, 2012*; *McKenna et al., 2010*). The calling algorithm for copy number variants (insertions or deletions longer than 100 bp) is described below. All SNVs, indels, and large deletions/duplications within the reportable range are analyzed and classified by the method described in the section "Variant classification." All reportable calls are reviewed by licensed clinical laboratory personnel.

## CNV calling algorithm

Our method for CNV calling is based on high-resolution depth of coverage analysis and performed in a manner similar to that successfully used by other groups (*Nord et al., 2011*; *Judkins et al., 2015*; *Lincoln et al., 2015*).

Analysis is performed on a per-lane basis. The region of interest for the assay is grouped into a number of regions for which copy number is counted (e.g., exons); each exon is considered independently and with no smoothing (e.g., HMM). Define matrix $d_{i,j}$ to be the matrix containing the number of reads from sample $i$ overlapping with region $j$. This matrix must be normalized. To protect against normalization issues due to individual samples with very large CNVs (such as a whole-gene deletion), we generate a normalization matrix $n_{i,j}$ by removing the highest variance probes from the total data set $D$ via the invariant set method described in *Li & Hung Wong (2001)*. The data matrix $d$ is then normalized in two steps:

$$d'_{i,j} = d_{i,j}/\text{mean}(n_{i,j} \text{ for all } j)$$
$$d''_{i,j} = d'_{i,j}/\text{mean}(n_{i,j} \text{ for all } i).$$

For each putative CNV $j$ in sample $i$, a hypothetical copy number and corresponding $Z$-score is computed:

$$c_{i,j} = 2 * d''_{i,j}$$
$$z_{i,j} = (d''_{i,j} - \text{mean}(d''_{i,j} \text{ for all } i))/\text{stdev}(d''_{i,j} \text{ for all } i).$$

A CNV call is considered confidently non-reference if abs $(z) \geq 4$ and the estimate $c$ is <1.2 or >2.8.

## Assay quality metrics

To ensure the quality of the results obtained from the Counsyl Inherited Cancer Screen, documentation and QC systems (Table S3) were developed in the Counsyl CLIA (Clinical Laboratory Improvement Amendments)-certified laboratory. Ancillary quality-control metrics, including amount of DNA recovered from a specimen ($\geq$18 ng/ul), fraction of sample contamination (<5%), unreliable GC bias, read qualities (percent Q30 bases per Illumina specifications), depth of coverage (per base target coverage >20×), are computed on the final output and used to exclude and re-run failed samples.

## Variant classification

Variants are classified according to the ACMG Standards and Guidelines for the Interpretation of Sequence Variants (*American College of Medical Genetics and Genomics, 2015*). All variants that are known or predicted to be pathogenic are reported; patients and providers have an option to have variants of uncertain significance reported as well. Final variant classifications are regularly uploaded to ClinVar.

## Statistical analysis

Validation metrics were defined as: Accuracy = (TP + TN)/(TP + FP + TN + FN); Sensitivity = TP/(TP + FN); Specificity = TN/(TN+FP); FDR = FP/(TP + FP), where TP = true positives, TN = true negatives, FP = false positives, FN = false negatives, and FDR = false discovery rate. The confidence intervals (CIs) were calculated by the method of *Wilson (1927)*.

# RESULTS

## Assay development

The Counsyl Inherited Cancer Screen we have developed employs next-generation sequencing and includes comprehensive analysis of all coding exons of *BRCA1* and *BRCA2*, 20 bp of flanking intronic sequences, and selected intronic and untranslated regions with known pathogenic variants (Table S2). The test was designed and optimized to detect single-nucleotide variants, indels, and copy-number variants. A proprietary bioinformatics pipeline was developed for sequence data alignment and variant detection as described above. To test the analytical sensitivity, specificity and accuracy of the assay, we sequenced peripheral blood and cell line DNA from 114 samples with an extremely high mean read depth (more than 500×) across all samples. Every targeted position was covered with a minimum of 20 reads.

## Analytical validation

To establish analytical accuracy for detecting single-nucleotide variants and indels, we compared Counsyl *BRCA1/2* sequence data of 41 Coriell samples (listed in Table S1) to reference data obtained from the 1,000 Genomes project and Counsyl *BRCA1/2* sequence data for NA12878 to high-quality reference data published by Illumina, Inc.

**Table 3  Performance of Counsyl Inherited Cancer Screen for SNPs and indels.**

| Counsyl ICS | 1000 Genomes reference data | | Results (95% confidence interval) |
| --- | --- | --- | --- |
| | **Variant present** | **Variant not present** | |
| Variant detected | 536 true positives | 0 false positives | 100% accuracy (0.999–1.0) |
| Variant not detected | 0 false negatives | 12,920 true negatives | 100% sensitivity (0.993–1.0) |
| | | | 100% specificity (0.999–1.0) |
| | | | 0% FDR (0–0.7%) |

Notes.

Only samples with reference data for the entire region of interest were used to calculate the analytic concordance. Validation metrics were defined as: Accuracy = (TP + TN)/(TP + FP + TN + FN); Sensitivity = TP/(TP + FN); Specificity = TN/(TN + FP); FDR = FP/(TP + FP). For true negative and true positive calculations, all polymorphic positions (positions at which we observed non-reference bases in any sample) across all samples were considered.

**Table 4  Positive variants included in validation study.**

| Mutation type | Subtype | Number of positive variants | |
| --- | --- | --- | --- |
| | | **Reference data** | **Orthogonal confirmation** |
| SNV | N/A | 525 | |
| Indel | Indels < 10 bp | 10 | 4 |
| | Indels ≥ 10 bp | 1 | 6 |
| CNV | Single-exon deletions or duplications | | 10 |
| | Multiple exon deletions or duplications | | 69 |

Notes.

Number of variants for the "multiple exon deletions and duplications" subtype is calculated by counting individual exons affected by the deletion/duplications.

(http://www.illumina.com/platinumgenomes/) (Table 2). The results presented in Table 3 demonstrate that 536 true positive calls, 12,920 true negative calls and no false positive or false negative calls were observed. In addition, to confirm the detection of documented variants in *BRCA1/BRCA2*, 15 samples from the BIC *BRCA1/BRCA2* Mutation Panel (available from Coriell) were included in the validation (Table 2 and Tables S1 and S4). The concordance between the *BRCA1/2* mutations detected by the Counsyl ICS and the BIC reference data was 100%.

Furthermore, to demonstrate the accurate detection of variants that are technically challenging for NGS, 10 pathogenic indels discovered in patient blood samples were subjected to orthogonal confirmation by Sanger sequencing (Tables 2 and 4; Table S4). All the indels detected by the Counsyl Inherited Cancer Screen were concordant with the Sanger results.

To establish analytical accuracy for detecting CNVs, we compared Counsyl copy number calls to CNV calls provided by reference labs, when available, and MLPA assays otherwise, on 63 samples: 13 samples from reference labs; 15 samples from the BIC *BRCA1/BRCA2* Mutation Panel; 25 random blood samples; 9 patient samples positive for CNVs; and NA12878 (Table 2). As shown in Table 5, 79 true positive calls, 3,067 true negative calls and no false positive or false negative calls were observed from the analysis of 63 analyzed samples (Table 5). Among the 63 tested samples, 10 had a deletion or duplication of a single exon, which can be technically challenging for NGS-based analysis.

**Table 5  Performance of Counsyl Inherited Cancer Screen for copy number variants.**

| Counsyl ICS | MLPA reference data | | Results (95% confidence interval) |
|---|---|---|---|
| | **CNV present** | **CNV not present** | |
| CNV detected | 79 true positives | 0 false positives | 100% accuracy (0.999–1.0) <br> 100% sensitivity (0.951–1.0) |
| CNV not detected | 0 false negatives | 3,067 true negatives | 100% specificity (0.999–1.0) <br> 0% FDR (0–4.6%) |

**Notes.**
Only reference data with full copy number assessment of the *BRCA1/2* genes were included. Validation metrics were defined as: Accuracy = (TP + TN)/(TP + FP + TN + FN); Sensitivity = TP/(TP + FN); Specificity = TN/(TN + FP); FDR = FP/(TP + FP). True positives and true negatives were computed on a per-exon basis.

The accuracy, sensitivity and specificity are therefore all 100% for SNPs, indels, and copy number variants. Only samples with reference data for the entire region of interest were used to calculate these metrics in order to avoid overestimating sensitivity or the confidence interval (*McAdam, 2000*). We also calculated false discovery rate (FDR) and the associated confidence interval (Tables 3 and 5). For SNPs and indels, the FDR is 0 of 536 positives, or 0% (95% CI [0–0.7%]). For CNVs, the FDR is 0 of 79, or 0% (95% CI [0–4.6%]).

Our validation samples represent a diversity of variant subtypes (Table 4 and Table S4), including 28 samples with variants technically challenging to detect by NGS, such as large indels and CNVs.

### Inter-run and intra-run reproducibility

In addition to establishing the test analytical sensitivity, specificity and accuracy, the Counsyl *BRCA1/2* test was validated for intra- and inter-run reproducibility. For indel detection reproducibility, each BIC sample ($n = 15$) was run three times each on three flow cells, for a total of nine replicates (Table S5). For SNV detection reproducibility, 11 deidentified blood samples were rerun on 2–3 different flow cells (Table S5). For CNV detection reproducibility, 15 Coriell cell line DNA and 11 patient samples were analyzed in replicates (Table S5). Concordance between replicates was 100% (Table S5), with no differences between inter- and intra-run replicates observed.

### Test compatibility with different input materials

Finally, to demonstrate compatibility with different sample types, deidentified paired blood and saliva samples (seven pairs) were tested. The results from paired blood and saliva samples were 100% concordant (Table S5).

## DISCUSSION

Pathogenic mutations in the *BRCA1/2* genes are known to be associated with increased risk for breast, ovarian and other cancers. For women, the risk of developing breast cancer by age 70 is approximately 60–70% for *BRCA1* and 45–55% for *BRCA2* mutation carriers. The cumulative ovarian cancer risk by age 70 (including fallopian tube and primary peritoneal carcinomas) is 40% for *BRCA1* and 20% for *BRCA2* mutation carriers respectively (*Antoniou et al., 2003*; *Chen & Parmigiani, 2007*; *King, Marks & Mandell, 2003*).

Increasing evidence indicates that early identification of BRCA carriers is important so that they can take advantage of genetic counseling, screening, and potentially life-saving prevention strategies.

NGS is increasingly being applied in the field of diagnostics, including *BRCA* analysis. An optimized and validated assay design is critical to maximizing the analytical performance of NGS assays and ensuring high-quality interpretation to facilitate clinical-decision making. Here, we describe the design and analytic validation of the Counsyl Inherited Cancer Screen, a next-generation-sequencing-based test to detect pathogenic variation in the *BRCA1*/2 genes. We demonstrate that the test is capable of detecting SNVs, indels, and copy-number variants with zero errors over a 114-sample validation set consisting of samples from cell lines and deidentified patient samples. Among the 114 tested samples, 28 (25%) were samples with challenging variants, including single- and multi-exon deletions/duplications ($n = 22$) and >10 bp indels ($n = 6$). The high sensitivity and specificity achieved in our study are comparable to the results of similar studies (*Bosdet et al.*, *2013*; *Judkins et al.*, *2015*; *Lincoln et al.*, *2015*; *Strom et al.*, *2015*), although some NGS studies report a higher false positive rate (*Chong et al.*, *2014*).

Some laboratories confirm NGS findings to reduce the risk of false positives (*Chong et al.*, *2014*; *Lincoln et al.*, *2015*). Result confirmation is recommended when the analytic false positive rate is high or not yet well established (*Rehm et al.*, *2013*). Confirmation can also be used to verify sample identity, which is critical when laboratory workflows are complex and not fully automated (*Rehm et al.*, *2013*). However, more recent work indicates that variant calls by NGS may be more reliable than relying on Sanger confirmation (*Beck et al.*, *2016*). In this study, we have orthogonally confirmed a number of SNV and CNV calls (using Sanger or MLPA testing, respectively: Table 2), and observed neither false positive nor false negative calls for sample with reference data across the entire region of interest. Additionally, the laboratory workflow described is fully automated, with positive sample tracking throughout the entire process. Barcode scans are performed of each tube and plate at all handling and pipetting steps. All sample processing actions performed by automated instruments are logged, and video of the decks of liquid handling instruments is continuously recorded.

In conclusion, we describe the development and analytical validation of a cost-effective, high-throughput NGS assay for the detection of *BRCA1* and *BRCA2* pathogenic mutations suitable for the clinical laboratory. We confirm that our test meets the rigorous quality standards necessary for clinical implementation (*Rehm et al.*, *2013*). The test is offered by Counsyl's laboratory, which is CLIA certified (05D1102604), CAP accredited (7519776), and NYS permitted (8535).

### Funding

The authors received no funding for this work.

## Competing Interests

All authors are employees and shareholders of Counsyl Inc., and the manuscript describes the validation of a commercial genetic test.

## Author Contributions

- Hyunseok P. Kang conceived and designed the experiments, analyzed the data, wrote the paper, prepared figures and/or tables, reviewed drafts of the paper.
- Jared R. Maguire analyzed the data, contributed reagents/materials/analysis tools, reviewed drafts of the paper.
- Clement S. Chu conceived and designed the experiments, performed the experiments, contributed reagents/materials/analysis tools, reviewed drafts of the paper.
- Imran S. Haque analyzed the data, contributed reagents/materials/analysis tools, wrote the paper, reviewed drafts of the paper.
- Henry Lai performed the experiments, contributed reagents/materials/analysis tools, reviewed drafts of the paper.
- Rebecca Mar-Heyming analyzed the data, reviewed drafts of the paper.
- Kaylene Ready conceived and designed the experiments, wrote the paper, prepared figures and/or tables, reviewed drafts of the paper.
- Valentina S. Vysotskaia wrote the paper, reviewed drafts of the paper.
- Eric A. Evans conceived and designed the experiments, analyzed the data, reviewed drafts of the paper.

## Human Ethics

The following information was supplied relating to ethical approvals (i.e., approving body and any reference numbers):

Western Institutional Review Board (IRB number 1145639).

## Data Availability

ClinVar (320494); http://www.ncbi.nlm.nih.gov/clinvar/submitters/320494/.

## Supplemental Information

Supplemental information for this article can be found online at http://dx.doi.org/10.7717/peerj.2162#supplemental-information.

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
