# Peer review of "Design and validation of a next generation sequencing assay for hereditary *BRCA1* and *BRCA2* mutation testing"

_PeerJ, doi:10.7717/peerj.2162_

## Round 0.1 · original submission · Major Revisions

· Academic Editor

Major Revisions

We could consider a substantial revision of the manuscript to address all the concerns of the three reviewers. In particular, please pay attention in your revision to validate your methods on a larger dataset, and to describe the details of the procedure in a way that is possible to reproduce it, as well as to make the dataset used freely available.

Reviewer 1 ·

Basic reporting

Kang et al. describe the clinical background and analytic validation of an NGS based test for germline sequence and copy number variants in BRCA1 and BRCA2 (BRCA1/2). Not all clinical laboratories publish or otherwise make their validation studies broadly available, and I find it commendable and important that Kang et al. have elected to do so. The authors’ study design, findings and reporting are generally sound and indeed should be published. However, certain issues must first be addressed in this manuscript prior to publication in PeerJ. In my estimation, these topics should not be difficult for the authors to address even though my description of them is somewhat lengthy (in an attempt to be clear).

1. Regarding the PeerJ requirement “The article should include sufficient introduction and background to demonstrate how the work fits into the broader field of knowledge”: The key finding in this manuscript is the validity of this lab’s NGS-based test for BRCA1/2. However the manuscript’s introduction and discussion sections do not address the history, guidelines or challenges/solutions in implementing clinical tests using NGS. The authors should include some text on this topic, which is often discussed in comparable papers.

2. Regarding the PeerJ requirement that “Relevant prior literature should be appropriately referenced”: Some comparable published validation studies of clinical NGS assays including BRCA1/2 are: Bosdet et al. JMD 2013; Chong et al. PLOS One 2014; Lincoln et al. JMD 2015; and Judkins et al. BMC Cancer 2015. Some other papers describe validation within a prevalence study, e.g. Maxwell et al., GIM 2014. These papers should probably be cited, and as relevant, compared/contrasted. Two specific areas where a comparison to one or more of these papers may be valuable are mentioned below in this review.

The ACMG NGS guidelines (Rehm et al. GIM 2015) also discuss validation of NGS based tests and should be cited. If the authors believe their study is not consistent with these guidelines it would be important to elaborate. The authors should mention if they have achieved CLIA, CAP or other regulatory approvals and cite those requirements.

3. Regarding PeerJ’s data sharing policy, it is not clear exactly how a reader can get the key data from this study:
- The specific list of Coriell and BIC samples used
- The specific variants observed in all samples in the study. Standard clinical nomenclature (HGVS) and clinical classifications (pathogenicity) should be provided. VCF format data may be useful in addition but is not required.
- The evidence underlying the clinical classifications of these variants
PeerJ’s policy requires “raw” data. This reviewer believes that lower level NGS data (BAM, FASTQ) could be valuable but is burdensome and unlikely to be used by a reader.

4. Tables 3 and 4 would be clearer to some readers if they used the standard 2x2 format for TP, FP, TN, and FN, and if rates were given as percentages.

5. As the manuscript discusses BRCA1/2, and not the broader hereditary cancer panel offered by the authors’ laboratory, the title of this paper will be misleading to some readers. There is variability in how the term “hereditary breast and ovarian cancer” (HBOC) is used today: technically it refers to only BRCA1/2 related cancers in some contexts, although increasingly in peer reviewed literature HBOC is used to describe BRCA1/2 and related genes, whether that usage is technically correct or not. This reviewer suggests the title “Design and validation of a next generation sequencing assay for BRCA1 and BRCA2”, as per the similar paper by Bosdet et al. (JMD 2013), will be far clearer today and in the future.

Experimental design

1. Re: Methods: Variant Classification. The ACMG 2015 ISV guidelines, cited in the methods, recommend labs classify variants as “pathogenic” (putatively disease causing), not “deleterious” (which sometimes implies only a molecular phenotype). Evidence for links to disease are sometimes indirect but the authors should clarify their reporting criteria and terminology compared to the referenced guideline.

2. In the methods the authors state that “Variant classifications are uploaded to ClinVar”. Currently no BRCA1/2 variants submitted by the authors’ laboratory appear to be in ClinVar. The authors should clarify.

3. While the methods describe variant classification, it is not clear how that played a role in the data presented here. This manuscript surrounds a validation study, and if the authors describe a method they should show data validating that method. For example, comparisons to interpretations from another laboratory, even though not a “gold standard”, are informative. Such a comparison could utilize the variants in tables 3 and 4, or a different representative set of variants. Alternatively the authors could limit the manuscript to analytic validation and clinical background and remove discussion of classification.

4. The “Test Description” in the methods section is appropriate for laboratory marketing literature, but probably not scientific manuscripts. Some more detail should be provided: e.g. which hybridization method is used, what NGS read-length and insert size (if paired ends), which of the GATK calling algorithms, and what specific QC thresholds must be met. Specific reference transcripts for BRCA1 and BRCA2 should be named, per HGVS standards. The authors mention that “selected intronic regions” are tested: which ones? Some of this information could be supplemental.

5. Important aspects of the validation are not explicitly described and will not be obvious to many readers:
a- What statistical method was used to calculate the CIs? Some CI methods are not appropriate for small N’s or error rates at or near zero.
b- Were all variants in BRCA1/2 in all samples compared, or only those with certain clinical interpretations? Does this vary by sample?
c- On what basis were TNs calculated? Per base pair? Per exon? Per gene? For only known polymorphic sites?

Validity of the findings

1. Re: Analysis and Statistics (tables 3 and 4):

a- Clearly CIs are not estimated to 7-digit precision in this study.

b- Analytic PPV (or equivalently, FDR) can be a more informative metric than specificity or FPR for such tests and should be included

c- As indels tend to be much more difficult to detect than SNVs, they should be broken out separately in table 3. Larger indels (say, >10bp) tend to be harder than smaller ones and should be broken out or listed separately, even though the number of large events is likely to be small in this study.

d- Similarly for CNVs, small events (single exon) tend to be harder that large CNVs (multi-exon). Duplications (CN=3) can be harder than deletions (CN=1). These too should be broken out or listed.

e- Some of the reference data in this validation study are from orthogonal confirmation. I assume this means that samples had an event, detected by the authors’ NGS method, which was then confirmed using another technique. Obviously such data can indicate little about sensitivity -- you have no idea what you missed. The authors should clarify and, depending on the answer, should ensure that sensitivity is calculated using only appropriate data points.

f- Without seeing the list, it is not possible to know whether the population of variants in this study is representative of those that would be reportable in a typical patient population. Some studies (Lincoln et al. JMD 2015, and others) have shown that technically difficult variants make up a disproportionate fraction of clinically relevant findings in hereditary cancer genes. A similar result has been presented by one of this manuscript’s authors (J. Maguire) in a recent talk (AMP 2015). The authors should briefly address.

2a. Some other validation studies (e.g. Chong et al. PLOS One 2014) report much higher false positive rates from clinical NGS than Kang et al. observe, although other studies (e.g. Lincoln et al. JMD 2015) show no FPs yet high sensitivity, like Kang. The authors may wish to address the reason behind their low FP results in the discussion, and how that came at no apparent cost to sensitivity.

2b. Per the ACMG NGS guidelines, to reduce FPs laboratories should perform confirmatory testing (e.g. by Sanger sequencing) until they have amassed sufficient data and experience to demonstrate that confirmation is not needed. The authors should describe their confirmation procedures, if any, and should describe data that informed that decision.

Comments for the author

Transparency is critical in the clinical diagnostics field. I commend Kang et al. for submitting this manuscript for peer review. While not yet ready for publication, I am optimistic that the issues in it can be quickly addressed and an excellent publication will emerge.

My greatest comment with the manuscript overall is that it reads like two different papers joined together. Most of the (lengthy) introduction and discussion text are essentially a review article, not original research. The (short) validation results are original research. As BRCA1/2 testing is well accepted and written into medical guidelines, some of this introductory text may be longer than is required to describe “design and validation” of such a test. Alternatively the authors could re-cast their paper more explicitly as addressing the validity of their BRCA1/2 test in the ACCE framework and keep the lengthy review text.

I should note that I am not a clinician, and I am not fully qualified to review all of the introduction and discussion text. From my non-expert knowledge it all seems both correct and comprehensive. I had no specific critiques, other than perhaps length.

I did not declare a conflict of interest in this review but I fully disclose that I work for another laboratory that offers clinical genetic testing services. In clinical practice, labs often review each other (e.g. CAP inspections and proficiency tests), and we do so with as objective and fact based an approach as possible. I have made every attempt to do that here.

Reviewer 2 ·

Basic reporting

No Comments

Experimental design

See the comments to authors

Validity of the findings

It is impossible to state the validity and robustness of the reported data as the description of the method has serious weaknesses and there is no possibility to access the data.

Comments for the author

The paper by Kang et al. reports the design and validation of a next-generation-sequencing-based test to detect pathogenic variation in the BRCA1 and BRCA2 genes. The introduction presents a well described motivation of the study. However the description of the proposed method has serious weaknesses. First the authors describe their CNV calling algorithm where a putative region is called CNV if it deviates from a baseline reference according to some predetermined threshold. The main problem in their method is that it relies on the underlying distribution of the data in the matrix d_{i,j}. The validity of this simplified cnv calling algorithm is by no way analysed and/or compared with current literature. An extensive evaluation on much larger dataset (there are many publicly available datasets which can be used for this purpose, e.g. the Conrad and the McCarroll datasets) and a comparison with the available tools is need. Also, it is not clear how a new sample can be analysed using this procedure.
Moreover, the main part of the method (Variant Classification) is simply outlined referring to some "custom curation software". There is no detail on how this classification works, where the corresponding data can be downloaded. and more important how their procedure can be replicated.
Overall the description seems the validation of a custom internal procedure, with no way of reproducing the results and no specific details (as for example the list of putative CNV) and no availability of the data used for the validation.
It is impossible to state the validity and robustness of the reported data as the description of the method has serious weaknesses and there is no possibility to access the data.
I recommend to validate the method on larger dataset, and to definitively describe the details of the test.

Reviewer 3 ·

Basic reporting

The clinical utility and validity of BRCA1 and BRCA2 is well described here. The authors present a comprehensive validation plan to detect multiple variant types in BRCA1 and BRCA2 genes. In light of recent discussion on proposed regulatory changes, these type of articles will demonstrate the rigorous testing clinical laboratories go through before introducing new diagnostic tests.

Experimental design

Validation of a new diagnostic test can present several challenged. This publication summarizes a validation strategy for detecting multiple variant types. Specific comments are listed below:
Specific comments are listed below:

1. List description of the validation samples that were used. In particular provide specific details about which BRCA1 and BRCA2 pathogenic mutations were specifically tested in the 15 deidentified samples. Where these known, highly prevalent mutations? What type of mutations were tested? Missense, splice site indels?
2. the 56 coriell samples for SNvs and 41 samples used for CNVs- were there particular samples with known mutations that were chosen? What type of copy number changes were analyzed? Long indels, single exon deletions, duplications?
2. A key performance metric for validating a new assay is demonstrated robustness. Provide data demonstrating the precision (intra run) and reproducibility (inter-run).
3. Performance metrics can vary by variant type. Describe sensitivity and specificity for SNVs, and indels separately.

Validity of the findings

See above:

---

## Round 0.2 · Minor Revisions

· Academic Editor

Minor Revisions

The manuscript has been remarkably improved but there are still minor comments by reviewer #1 that is important to address. I would be glad to consider a further revised version of the manuscript in light of the precious criticisms raised by the reviewer.

Reviewer 1 ·

Basic reporting

Kang et al.’s revised manuscript is a significant improvement over the original draft: It is now clear, thorough, and readable. The study appears valid and worthy of publication, which I recommend subject to minor revisions noted here. I believe the authors have addressed well the most important comments that I had previously mentioned.

The manuscript meets PeerJ’s standards for basic reporting. Data are appropriately disclosed. A few additional citations would be appropriate (mentioned below).

Experimental design

The manuscript meets PeerJ’s standards for experimental design. Methods are appropriately disclosed, with these minor exceptions that I believe should be briefly addressed:

1. Re: Line 123-144. The CNV method is reminiscent of other read-depth based approaches, which should be stated and cited. Nord et al (BMC Genomics 2011) describe the first (I believe) clinical NGS CNV approach for BRCA1/2, and both the Judkins and Lincoln papers used an NGS based CNV approach, in addition to various research papers. As there is skepticism in the clinical community about NGS based CNV detection, citations will help justify the author’s choice (in addition to the data they present).

2. Unlike some other NGS CNV techniques, the author’s CNV method does not seem to use any smoothing or other inference between neighboring “regions” (e.g. an HMM or a similar algorithm). This probably should be stated explicitly.

Without such an algorithm (HMM, etc.) the counting by exon in table 5 is independent and thus valid. Indeed a phrase could be added to justify this analysis, as to some readers the justification will be non-obvious. However if there IS such an algorithm then it is NOT appropriate to count per exon (the tests are not independent) and the metric should be changed to per CNV counting.

3. Re: Lines 103-111. The specific hybridization method or probe type should be named.

4. Re: Lines 146-153. The description of QC is a bit vague. The authors should include (perhaps as a supplemental table) all specific QC criteria that allow them to achieve the excellent performance observed in this study. Metrics related to CNV calling should be included.

5. Re: Line 257-266. The authors should describe what “positive sample tracking” is used.

Validity of the findings

The results appear valid. I believe one text clarification and two omissions should be addressed.

5. The author’s statement (line 210-211, and table 3 legend) is important: Unlike a mistake made in some other papers, these authors did NOT use targeted orthogonal confirmation data to measure sensitivity. As a learning exercise for readers, the authors may wish to clarify why this restriction in lines 210-211 was imposed, e.g. adding “…in order to avoid overestimating sensitivity or its confidence interval”. The authors may find MacAdam (J Clin Microbiol, 2000) an appropriate reference for such a statement.

6. A related data point is however missing in this manuscript: It is important to know the breakdown of SNVs vs. small indels vs. big indels in the TP count in table 3. One possible way to accomplish this would be to make table 4 more inclusive, also including a row for SNVs, and providing another column so that the count of each variant type could be broken down by those samples that had complete independent assessment, vs. those that were subject only to orthogonal confirmation. Another column could be the total count if desired.

7. The most significant omission is in the discussion of confirmation (lines 257-266). While the author's data shows no FPs, the risk of a FP is not quantified in this text but should be to justify the statements made. Specificity is not an appropriate or intuitive measure for this, rather FDR (false discovery rate) and the associated CI is. FDR in this usage is the fraction of reported positives that are not positive: FP / (TP + FP). For sequence variants the author's data shows 0 of 521 or 0% FDR with a Wilson CI of 0 to 0.73%, calculated on a per variant basis and not conditioned on prevalence. Note that the N may be larger than 521 as the study samples with only orthogonal confirmation ARE appropriate to include in this particular calculation. CNVs should be similarly mentioned. Alternatively the authors could perform a more complex per-patient FDR calculation incorporating the prevalence of variants in BRCA1/2. Analytic PPV is 1 - FDR and may be used if preferred, but note that PPV is generally reported conditioned on prevalence.

Comments for the author

The authors have made excellent progress on describing their study.

Reviewer 2 ·

Basic reporting

No Comments

Experimental design

No Comments

Validity of the findings

No Comments

Reviewer 3 ·

Basic reporting

The revised article contains information on introduction and background. It reports the design and validation of a next-generation-sequencing-based test to detect pathogenic variation in the BRCA1 and BRCA2 genes. The introduction and background are clearly defined.

Experimental design

The revised articile is now limited to analytical validation. The data set is much more clearly defined with adequate information on controls used and methodologies, and the spectrum of mutations tested for validation.

Validity of the findings

The revised dataset conforms to the compliance and regulatory standards and provides information about laboratory validation process for BRCA1 and BRCA2.

---

## Round 0.3 · accepted · Accept

· Academic Editor

Accept

The revised manuscript nicely addresses all the previous concerns and can be now endorsed for publication.